# Desert dust exerts twice the longwave radiative heating estimated by climate models

Jasper F. Kok [1] ✉, Ashok K. Gupta[1,2], Amato T. Evan [3], Carlos Pérez García-Pando [4,5], Longlei Li [6], Adeyemi A. Adebiyi [7], Samuel Albani [8], Yves Balkanski[9], Ramiro Checa-Garcia[9,10], Peter R. Colarco [11], Douglas S. Hamilton [12], Yue Huang[1], Akinori Ito [13], Martina Klose[14], Natalie M. Mahowald [6], Ron L. Miller[15], Vincenzo Obiso[4], Adriana Rocha Lima[16] & Jessica Wan [6,17]

Although desert dust is the most abundant atmospheric aerosol by mass, its longwave radiative effects remain unclear, obscuring the impacts of dust on weather and climate. Here, using a data-driven analytical model constrained by observations, we show that scattering and absorption of longwave radiation by dust heats the planet by $+0.25 \pm 0.06$ W m$^{-2}$ (90% confidence). This is nearly twice the value simulated by current climate models, which omit longwave scattering and underrepresent super coarse dust (diameter > 10 μm). These omissions bias modeled surface energy fluxes, cloud responses, precipitation, and atmospheric circulation. At the global scale, the sign and magnitude of the net dust direct radiative effect remain uncertain, with additional work needed to constrain shortwave cooling effects. These findings show that improving the representation of dust interactions with longwave radiation can improve weather forecasting and is essential to resolve the role of dust in climate change.

Desert dust is the most abundant aerosol by mass in Earth's atmosphere[1,2] and exerts widespread influences on weather and climate through its interactions with radiation, atmospheric chemistry, clouds, precipitation, and atmospheric circulation[3–6]. Although the effects of dust on shortwave (SW) radiation have been extensively studied[7–9], its interactions with longwave (LW) radiation remain comparatively uncertain[4,10–12]. This uncertainty hampers assessments of dust's impact on weather and regional climate[3,13], and obscures whether historical increases in dust have acted to amplify or offset anthropogenic greenhouse warming[4].

[1]Department of Atmospheric and Oceanic Sciences, University of California, Los Angeles, CA, USA. [2]Department of Earth and Environmental Sciences, Vanderbilt University, Nashville, TN, USA. [3]Scripps Institution of Oceanography, University of California, San Diego, CA, USA. [4]Barcelona Supercomputing Center (BSC), Barcelona, Spain. [5]ICREA, Catalan Institution for Research and Advanced Studies, Barcelona, Spain. [6]Department of Earth and Atmospheric Sciences, Cornell University, Ithaca, NY, USA. [7]Department of Life and Environmental Sciences, University of California, Merced, CA, USA. [8]Department of Environmental and Earth Sciences, University of Milano-Bicocca, Milano, Italy. [9]Laboratoire des Sciences du Climat et de l'Environnement, CEA-CNRS-UVSQ-UPSaclay, Gif-sur-Yvette, France. [10]Royal Netherlands Meteorological Institute, De Bilt, the Netherlands. [11]Atmospheric Chemistry and Dynamics Laboratory, NASA Goddard Space Flight Center, Greenbelt, MD, USA. [12]Marine, Earth, and Atmospheric Science, North Carolina State University, Raleigh, NC, USA. [13]Yokohama Institute for Earth Sciences, JAMSTEC, Yokohama, Kanagawa, Japan. [14]Institute of Meteorology and Climate Research Troposphere Research, Karlsruhe Institute of Technology (KIT), Karlsruhe, Germany. [15]NASA Goddard Institute for Space Studies, New York, USA. [16]Department of Physics, University of Maryland, Baltimore County, 1000 Hilltop Circle, Baltimore, MD, USA. [17]Climate Systems Engineering initiative, University of Chicago, Chicago, IL, USA. ✉e-mail: jfkok@ucla.edu

Dust alters LW radiative fluxes by scattering and absorbing thermal radiation emitted by the atmosphere and the surface[7,9,11,12]. Dust itself normally emits LW radiation to space at a lower brightness temperature because it is situated higher in the atmosphere, causing net radiative cooling of the atmosphere and net heating at the surface and the top-of-atmosphere (TOA)[3,10]. These effects are most important for radiation with wavelengths in the "atmospheric window" around 8 – 14 μm in which a cloud-free atmosphere is relatively transparent. Outside of this window, the atmosphere is opaque to radiation due to absorption by water vapor and other greenhouse gases, such that the addition of another source of extinction has a negligible effect on the TOA spectral flux[10].

Climate model calculations of the resulting perturbation to Earth's global energy balance - the LW direct radiative effect (DRE) - range between approximately 0.1 to 0.25 Wm[-2][7,12] (Supplementary Table 1). This large uncertainty is a consequence of substantial biases and uncertainties in model simulations, including in the LW optical properties[14], altitude[15,16], and size distribution[17] of dust. For instance, many models underestimate, or even omit, the contribution of super coarse dust (with diameter $10 < D \leq 62.5$ μm), which might account for up to a third of the LW DRE[18]. In addition, most radiative transfer schemes used in global climate models do not account for the scattering of LW radiation[10,19], which previous work suggests accounts for approximately half of the total LW DRE[10,12,20]. It is therefore likely that climate models underestimate the LW DRE[10–12].

Since climate models are thus poorly suited to determine the heating effect due to dust interactions with LW radiation, here we quantify the dust LW DRE using a data-driven analytical model. Our model accounts for LW scattering and integrates observational constraints on dust optical properties, abundance (including super coarse dust), and LW radiative effects. Relative to climate model results, this observationally constrained analytical approach substantially reduces uncertainties and biases, yielding a global mean LW DRE at TOA of $0.25 \pm 0.06$ W/m$^2$ (90% confidence interval), which is approximately twice the 0.13 (0.09 – 0.23) Wm$^{-2}$ simulated by climate models (Supplementary Table 1). This model underestimation of the LW DRE has important implications for weather and climate.

## Results

### Data-driven analytical model of dust longwave direct radiative effects

In developing an analytical model for the dust LW DRE, we considered the key factors influencing the TOA LW DRE (see Methods and Supplementary Fig. 1). First, the LW DRE scales approximately linearly with the atmospheric abundance of dust[20], which is best constrained by remote sensing data of dust aerosol optical depth (DAOD) in the shortwave (SW) spectrum[21]. Second, the LW DRE depends on the highly uncertain[14] optical properties of dust (mass extinction efficiency, single-scattering albedo, and asymmetry parameter) in the LW spectrum. Third, the dust size distribution is critical, because coarse dust ($2.5 \leq D < 10$ μm) and super coarse dust likely contribute over 80% of the LW DAOD[1,18] (Supplementary Table 2). Fourth, the dust LW DRE generally increases with dust layer height[20,22]: elevated dust layers are generally colder, and thus emit at a lower brightness temperature than the surface. This reduces outgoing radiation to space, thereby enhancing the dust LW DRE. Fifth, the dust LW DRE decreases with greater absorptivity in the atmospheric window between the dust layer and the surface, as enhanced atmospheric absorption reduces the LW radiative flux reaching the dust layer. Finally, the dust LW DRE also decreases with the atmospheric absorptivity above the dust layer because absorption or downward scattering of upwelling LW radiation by dust will have no influence on the TOA energy budget if that upwelling LW radiation would have been absorbed by overlying greenhouse gases or clouds anyways. Consequently, optically thick clouds (i.e., with LW optical depth >> 1) overlying a dust layer suppress the dust LW DRE, whereas clouds beneath the dust layer only slightly reduce the LW DRE[10,20].

To capture the combined influence of these factors on the LW DRE, we use a data-driven analytical model (Supplementary Figs. 1, 2) that quantifies the effect of dust on the TOA LW radiative flux under clear-sky conditions. We drive this model with atmospheric and surface properties from reanalysis meteorology[23], the dust refractive index sampled from various observational studies[14], and joint observational-modeling constraints of the size and abundance of dust with diameters up to 100 μm from the DustCOMM data set[1]. To represent uncertainty, we propagate the various experimental, observational, and modeling uncertainties in these data sets through a bootstrap procedure, generating 1000 realizations of the LW DRE. We then apply observational estimates of the top-of-atmosphere LW DRE per unit of shortwave (∼550 nm) optical depth in clear-sky conditions[22,24] - the LW DRE efficiency (DREE) - to discard the subset (∼50%) of bootstrap realizations that are statistically inconsistent with these observations. The closure we thus achieve between the "bottom-up" calculation of the LW DRE by the analytical model and the "top-down" constraints from in situ and satellite data (Supplementary Fig. 3) provides confidence in our results.

### The spatiotemporal pattern of the LW direct radiative effect efficiency (DREE)

We find that our analytical model can reproduce, within uncertainties, the spatiotemporal patterns of satellite observations of the LW DREE (Figs. 1, 2a). Both model and satellite observations indicate that the LW DREE peaks in summer, when the temperature difference between the surface and the dust layer is largest (Supplementary Fig. 4), reflecting both high surface temperatures and stronger convection carrying dust to higher altitudes (Supplementary Fig. 5). We further find that the LW DREE is largest near source regions, where the particle size distribution is coarsest, resulting in greater LW radiative effects per unit SW DAOD (Supplementary Table 2). This enhancement is particularly pronounced in more isolated source regions such as Australia and East Asia, where dust is predominantly locally emitted and is thus relatively coarse. The LW DREE generally decreases away from source regions, as temperatures over oceans and vegetated regions are lower than over deserts and as gravitational settling preferentially removes coarse particles during transport, yielding finer dust[18].

The analytical model shows substantially better agreement with LW DREE observations than six different global model simulations. It largely matches the magnitude of the LW DREE observations (bias of −1.0 Wm$^2$), explains slightly less than half of the variance in the observations ($R^2 = 45\%$), and is statistically consistent with those observations (reduced chi square value of $\chi^2_\nu = 0.59$) (Fig. 2a). In contrast, global model simulations underestimate the LW DREE by approximately a factor of 2 and, on average, explain less than a quarter of the variance in observations (average $R^2 = 23\%$; Fig. 2b and Supplementary Table 3).

### Observationally constrained annual global mean LW DRE

We obtained the annual global mean clear-sky LW DRE at TOA by integrating our results over space and time, obtaining $0.32 \pm 0.08$ Wm$^{-2}$ (90% CI), which is consistent with recent radiative transfer modeling results[25]. Given that the dust SW DAOD is $0.03 \pm 0.01$[1,18,21,26], this result implies a global mean clear-sky LW DREE of $11 \pm 5$ Wm$^{-2}\tau_{SW}^{-1}$.

To estimate the more climatically relevant all-sky LW DRE, we applied ratios of all-sky to clear-sky LW DRE from an ensemble of global model simulations[27] (see "Methods" and Supplementary Figs. 2, 6). We find that the annual mean all-sky LW DRE (Fig. 3a) is of the order of several Wm$^{-2}$ close to the dust source regions, where LW DAOD (Fig. 3b) is largest. Because arid source regions have sparse cloud cover, clouds reduce the TOA LW DRE by only ∼10–20% near

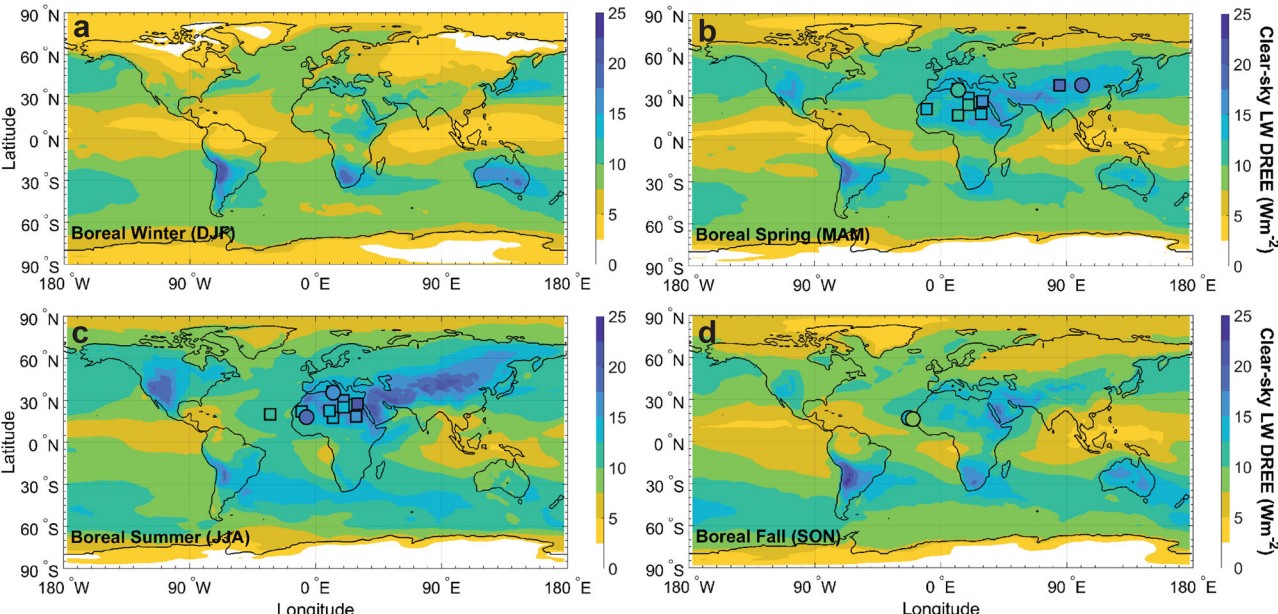

**Fig. 1 | Spatial and seasonal patterns of the top-of-atmosphere clear-sky long-wave (LW) direct radiative effect efficiency (DREE).** Panels show boreal winter (DJF; **a**), spring (MAM; **b**), summer (JJA; **c**), and fall (SON; **d**). Predictions from the data-driven analytical model reproduce the magnitude and variability of observational estimates based primarily on in situ measurements (colored circles) and satellite data (colored squares). Both model and observations indicate a range of the clear-sky LW DREE of ~ 5–20 Wm$^{-2}\tau_{SW}^{-1}$. The LW DREE is largest near dust source regions (primarily deserts), where the dust size distribution is coarsest, and during spring (**b**) and summer (**c**), when the surface is warmest, and the dust is elevated by stronger convection[49]. All panels represent the diurnally and seasonally averaged LW DREE (see "Methods").

most source regions, but by ~ 40–60% over remote regions such as oceans, which experience substantially higher cloud cover (Fig. 3c). This anti-correlation between dust and cloud cover causes cloud-induced reductions in LW DRE to be much smaller than the factor-of-two reductions typically seen for the SW radiative effect of other aerosols[28]. Overall, accounting for the effects of clouds on the LW DRE at TOA reduces the global DRE by ~ 20%, yielding an all-sky LW DRE of 0.25 ± 0.06 Wm$^{-2}$ (median and 90% confidence interval; Fig. 4).

## Discussion
### Why global models underestimate the LW DRE
A compilation of 24 global model results shows a median all-sky LW DRE of 0.13 (90% confidence interval: 0.09–0.23) Wm$^{-2}$ (Supplementary Table 1) – roughly half of our observationally constrained estimate (Fig. 4). In addition to underestimating the global mean LW DRE, models poorly capture the spatiotemporal variance of LW direct radiative effects (Fig. 2b and Supplementary Table 3).

A critical contributor to this model bias is the widespread neglect of LW scattering, which is not included in most radiative transfer schemes used in global models[29]. This omission is problematic because scattering accounts for approximately half of the LW DAOD [51% (16 to 60)] and an even larger fraction of the LW DRE [57 (21 to 66) %] (Fig. 4). Scattering is thus somewhat more effective than absorption in generating a top-of-atmosphere radiative effect; indeed, we find that the global LW DREs generated per unit LW DAOD due to scattering and due to absorption are 33 ± 7 and 26 ± 7 Wm$^{-2}\tau_{LW}^{-1}$, respectively (Supplementary Fig. 7). This higher efficiency of scattering in reducing the outgoing longwave radiation (OLR) occurs because 28 ± 3 % of scattering interactions result in down-scattering (Supplementary Fig. 8), which reduces OLR. In contrast, because a material's absorptivity scales both its absorption and its emission of LW radiation, per Kirchhoff's law, the effect of dust absorption of LW radiation is tempered by co-occurring emission of LW radiation, albeit at a lower temperature.

Another major contributor to the model underestimation of the LW DRE is the underestimation of coarse and super coarse dust[11,12,17]. The ability of dust to interact with LW radiation in the atmospheric window increases with particle diameter[30], with coarse (2.5 ≤ D < 10 μm) and super coarse (10 ≤ D ≤ 62.5 μm) dust respectively accounting for approximately 60 and 25% of the LW extinction[18] (Supplementary Table 2). Correspondingly, a large fraction of the LW DRE is produced by coarse (~ 65%) and super coarse (~ 20%) dust (Fig. 4 and Supplementary Table 2). However, many models substantially underestimate the concentration of coarse and (especially) super coarse dust[17,31,32], with many even omitting super coarse dust altogether[1], leading to further underestimation of the LW DRE (Fig. 4).

Addressing these model biases requires both remedying the underestimation of super coarse dust[17] and accounting for the scattering of LW radiation[10]. Recent parameterizations that account for the observation that the emission flux of super coarse dust is much greater than most models simulate provide a path forward[31,33–35]. However, models also appear to underestimate the lifetime of super coarse dust, such that improved treatments of the effects of turbulence[36], dust asphericity[37], dust orientation[38], and small-scale convection[39] on dust settling might be needed to reconcile model predictions with observations of super coarse dust far from source regions[18].

Models also need to account for the contribution of scattering to the LW DRE. Some modeling studies have tried to do so by simply scaling up the radiative effect due to LW absorption[7,11,12,19], but this approach neglects the spatiotemporal variability of the fractional contribution of scattering to the LW DRE (Supplementary Fig. 9). This variability arises primarily because the radiative perturbation from LW absorption depends strongly on the temperature contrast between the dust layer and the surface: absorbed radiation is re-emitted at the dust layer's colder temperature rather than the warmer surface temperature, leading to a net reduction in outgoing longwave radiation. In contrast, the radiative effect from LW scattering is largely independent of temperature contrast and thus of dust layer altitude. Consequently, the relative contribution of LW scattering to the LW DREE and thus the

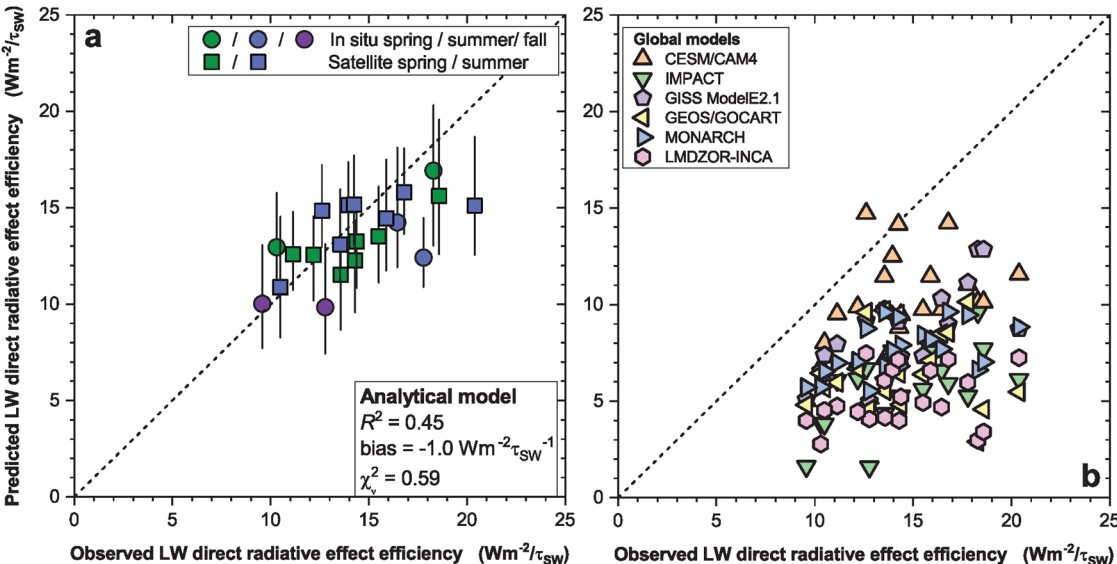

**Fig. 2 | Comparison between model predictions and observational estimates of clear-sky longwave (LW) direct radiative effect efficiency (DREE). a** The data-driven analytical model reproduces both the magnitude (bias = -1.0 $Wm^{-2}\tau_{SW}^{-1}$) and the seasonal and spatial variability ($R^2 = 0.45$) of observational estimates, agreeing with most observational estimates within the uncertainties (reduced chi squared $\chi_\nu^2 = 0.59$). **b** In contrast, global model simulations substantially underestimate the LW DREE observations, with a bias of − 4 to − 9 $Wm^{-2}\tau_{SW}^{-1}$ (Supplementary Table 3). This low bias likely occurs because models neglect LW scattering and under-estimate super coarse dust. Vertical error bars represent 90% confidence intervals; horizontal error bars are not shown to avoid cluttering the figure but are assumed to equal 2 $Wm^{-2}\tau_{SW}^{-1}$ (see Methods). The data shown represents the diurnally and seasonally averaged LW DREE (see "Methods").

DRE (Supplementary Fig. 9) is greater when the temperature difference between dust and the surface is smaller (Supplementary Fig. 4), which typically occurs when dust is lower in the atmosphere (Supplementary Fig. 5). This causes LW scattering to make a greater contribution to the LW DREE in winter than in summer and close to source regions than far from source regions (Supplementary Fig. 9).

Despite these differences in the spatiotemporal pattern of the radiative effects of LW absorption and scattering, we find that neglecting LW scattering by global models causes only modest errors in the spatiotemporal pattern of the dust LW DREE (Supplementary Fig. 10, left column). Thus, a simple approximation to account for the effects of LW scattering on the TOA energy budget is to use a mass extinction efficiency reflecting the contributions of both scattering and absorption, but to set the LW single-scattering albedo to zero for perfectly absorbing particles[19]. This approach degrades agreement against LW DREE observations but yields a central LW DRE estimate of 0.22 $Wm^{-2}$ (Supplementary Fig. 10, right column), which is close to our observationally based constraint. However, because absorption and scattering differ in their dynamical impacts—absorption can alter atmospheric stability through radiative cooling, whereas scattering does not[10] - it is preferable to represent dust LW scattering in radiative transfer schemes used in regional and global models. Machine learning parameterizations may offer a computationally efficient path forward[40].

**Implications of the missing LW radiative effects for weather and climate**

Our finding that models underestimate dust–longwave interactions by roughly a factor of two has important implications for dust impacts on weather and regional climate through its coupling with clouds, precipitation, and the surface energy balance. This model underestimation—driven primarily by the omission of LW scattering (Fig. 4)—leads to biases in both surface and top-of-atmosphere LW radiative effects. Although the atmospheric response to these radiative perturbations is mediated by interactions with the large-scale circulation[3], the net effect of underestimating dust LW radiative effects – whose magnitude varies only weakly over the diurnal cycle - is to overestimate dust's daytime surface cooling while underestimating its nighttime surface heating. This bias in the diurnal cycle can cause an overestimation of the negative feedback of dust radiative effects onto dust emission itself[41], as well as an underestimation of surface evaporation over oceans and vegetated regions, with a corresponding low bias in precipitation[3,42]. Moreover, because absorption and emission of LW radiation by dust overlying stratocumulus clouds reduce cloud-top cooling and modulate boundary-layer stability, underestimating LW radiative effects biases the representation of dust semi-direct effects on stratocumulus clouds[43]. Specifically, models that omit LW scattering tend to over-estimate cloud-top cooling, thereby strengthening the inversion and artificially increasing low-level cloud cover[43]. Incorporating the missing LW radiative effects of mineral dust could therefore yield substantial improvements in numerical weather prediction models and climate models[3,13,41].

Our constraints on the LW DRE also provide a critical step toward resolving the sign and magnitude of the net dust direct radiative forcing – that is, the change in the dust direct radiative effect from pre-industrial to present-day - and its contribution to climate change. However, because the counterbalancing cooling from the SW DRE remains highly uncertain[7,11,12] - with a recent review[4] constraining it to − 0.40 ± 0.25 $Wm^{-2}$ - the sign of the net dust DRE (SW + LW) is still unclear. As such, it remains uncertain whether historical increases in dust loading have enhanced or opposed anthropogenic greenhouse warming of the global climate system[4]. Nonetheless, as climate models start implementing historical changes in dust emissions[44,45], it is critical that they correct the low bias in the LW DRE; otherwise, climate model simulations of dust radiative forcing will remain inaccurate.

Although LW heating and SW cooling thus counteract each other in the global energy budget, their relative magnitudes vary greatly by region. Specifically, heating from the LW DRE is largest near dust source regions, whereas cooling from the SW DRE is strongest over oceans and other low-albedo surfaces downwind of sources[25,46] (Fig. 3d). The resulting dipole pattern in the net dust DRE – with heating over source regions and strong cooling over oceans and

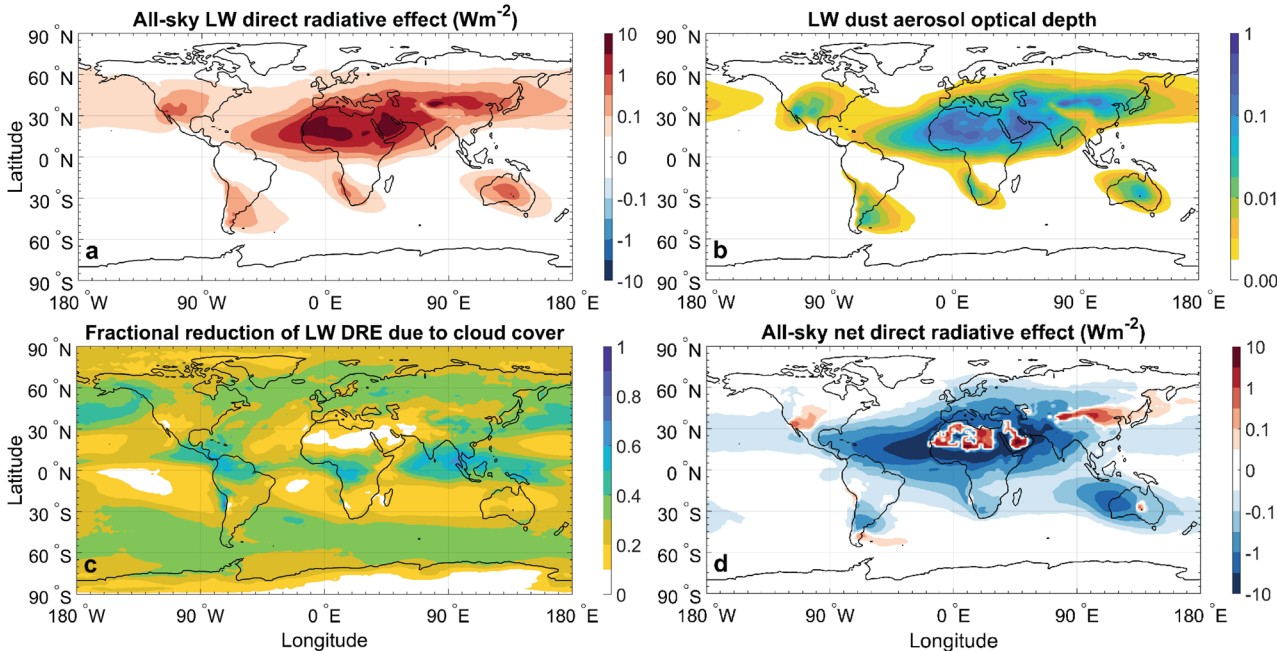

**Fig. 3 | Spatial pattern of the longwave (LW) direct radiative effect (DRE).** The top-of-atmosphere (TOA) LW DRE for all-sky conditions (**a**) is driven by extinction of LW radiation, as quantified by the LW dust aerosol optical depth (DAOD) (**b**). The LW DRE is mitigated by the effects of clouds (**c**) and can be offset by cooling from the shortwave (SW) DRE, resulting in a net DRE (**d**) that, on balance, heats over low-albedo dust source regions and strongly cools over downwind low-albedo regions such as oceans. The LW DRE and DAOD were obtained from our data-driven analytical model (see "Methods") and the fractional reduction of the LW DRE by clouds was obtained from an ensemble of global model simulations (see "Methods" and Supplementary Fig. 6). The net DRE was estimated by adding the SW DRE with observationally constrained dust SW optics from a recent study[79] to the LW DRE. All panels represent annually averaged results; the seasonally averaged LW DRE is shown in Supplementary Fig. 11.

vegetated regions near source regions - influences regional climate, atmospheric circulation, tropical cyclones, and monsoons[3,46–48].

In conclusion, we constrain the global annual mean heating from dust interactions with LW radiation to $+0.25 \pm 0.06$ W/m² at the top-of-atmosphere. This estimate is derived from a data-driven analytical model that integrates atmospheric and surface properties with observational constraints on dust properties and abundance. Our bottom-up calculations are statistically consistent with both the magnitude and spatial variability of observational estimates of LW direct radiative effects, yielding results in substantially better agreement with observations – and more tightly constrained - than possible with current climate models (Fig. 2). Most models neglect longwave scattering – responsible for more than half of the global mean LW DRE - and underestimate or omit super coarse dust, which contributes roughly 20%. Consequently, models underestimate the LW DRE by about a factor of two (Fig. 4), causing biases in surface energy fluxes, cloud semi-direct effects, and precipitation. The heating due to the dust LW DRE is partly counteracted by uncertain dust SW cooling, producing regional dipole patterns of radiative effects (Fig. 3d) important to regional climate and atmospheric circulation. Moreover, the uncertain SW cooling obscures whether dust exerts net global cooling or heating, and thus whether historical increases in dust have enhanced or offset anthropogenic greenhouse warming. Together, these findings highlight the critical need to better represent dust–longwave interactions in models to improve predictions of weather and regional climate and reduce fundamental uncertainties in the role of dust in Earth's energy balance and global climate change.

## Methods

### Data-driven analytical model of dust LW radiative effects at TOA

We use a combination of theory, model simulations, and observations of dust abundance and LW radiative effects to constrain the LW DRE at

TOA (Supplementary Figs. 1 and 2). First, we derive a simplified model that captures the essence of how the dust LW DRE at TOA depends on atmospheric, surface, and dust properties[10]. We then calculate the dust LW DRE by driving this model with data on dust properties and abundance from the DustCOMM data set (described in the Supplementary Methods and in refs. 1,17,49.), data on dust LW optical properties from various laboratory and in situ measurements[14], and data on atmospheric and surface properties from reanalysis data sets[50]. We propagate the uncertainties in these various data sets through a bootstrap procedure, yielding many (1000) simulations of the LW DRE. We then apply a compilation of observational estimates of the clear-sky LW DREE[22,24] to eliminate the subset of simulations (~55%) that are statistically inconsistent with these observations. By combining the clear-sky LW DRE with an ensemble of model simulations of the ratio of the all-sky to the clear-sky LW DRE, we then obtained the all-sky LW DRE at TOA.

As illustrated in Supplementary Fig. 1, we consider a dust layer with optical depth $\tau_{LW}$ at some LW wavelength $\lambda$ and effective emission temperature $T_d$ (see definition in Supplementary Methods and Supplementary Fig. 12e). The upwelling spectral flux (Wm⁻²μm⁻¹) immediately below the dust layer is[51]:

$$F_{s,\,eff\uparrow}(\lambda) = \pi B(T_{s,\,eff}, \lambda), \qquad (1)$$

where $B$ is the Planck function that describes the spectral intensity of a blackbody as a function of temperature. Further, $T_{s,\,eff}$ is the effective emission temperature of upwelling radiation below the dust layer, which is defined in the Supplementary Methods and depends primarily on the surface temperature $T_s$ (Supplementary Fig. 12a) and surface emissivity $\epsilon_s$ (Supplementary Fig. 12b), with a small correction due to absorption and emission by the atmosphere below the dust layer. Although the surface emissivity is close to 1 outside of source regions,

it can be substantially below 1 for desert regions (Supplementary Fig. 12b). This causes $T_{s,\text{eff}} - T_s$ to attain values of up to ~5 °C (Supplementary Fig. 12d), which acts to reduce dust longwave radiative effects.

The upwelling spectral irradiance immediately above the dust layer is then

$$F_{d\uparrow}(\lambda) = \pi B\left(T_{s,\text{eff}}, \lambda\right)\left[1 - \epsilon_d(\lambda)\right] + \epsilon_d(\lambda)\pi B(T_d, \lambda)$$
$$- \pi R_d(\lambda)\left[B\left(T_{s,\text{eff}}, \lambda\right) - \epsilon_{\text{abv}}(\lambda)B\left(T_{\text{abv}}, \lambda\right)\right] \quad (2)$$

where $\epsilon_d$ is the emissivity of the dust layer and $\epsilon_{\text{abv}}$ and $T_{\text{abv}}$ are respectively the absorptivity (Supplementary Fig. 12g) and effective emission temperature at wavelength $\lambda$ of the atmosphere above the dust layer, such that $\pi R_d \epsilon_{\text{abv}} B(T_{\text{abv}}, \lambda)$ represents the spectral radiance due to upward scattering by dust of radiation emitted downward by the overlying atmosphere. Equation (2) assumes that $\epsilon_d$ and $\epsilon_{\text{abv}}$ are « 1, which is a reasonable assumption only in the atmospheric window (see further discussion below). Further, $R_d$ is the fraction of upwelling radiation that is scattered downward by the dust layer. For isotropic radiation and in the limit of $\tau_{\text{LW}}$ « 1 (see Supplementary Methods for a discussion of the impact of this assumption), $\epsilon_d$ and $R_d$ equal[52,53]

$$\epsilon_d(\lambda) = [1 - \omega(\lambda)]\left[1 - \exp\left(-\frac{\tau_{\text{LW}}(\lambda)}{\widetilde{\mu}}\right)\right], \quad (3)$$

$$R_d(\lambda) = \omega(\lambda)\beta_\downarrow\left[1 - \exp\left(-\frac{\tau_{\text{LW}}(\lambda)}{\widetilde{\mu}}\right)\right], \quad (4)$$

where $\omega(\lambda)$ is the dust layer's single-scattering albedo (Supplementary Fig. 8) and $\widetilde{\mu} = 0.6$ is the cosine of the effective zenith angle[51]. Furthermore, $\beta_\downarrow$ is the downscatter fraction, that is, the fraction of dust-scattered upwelling LW radiation that is scattered back towards Earth's surface. Note that the downscatter fraction $\beta_\downarrow$ thus differs from the backscatter fraction $b$, which is the fraction scattered into the backward hemisphere relative to the direction of propagation of the incoming radiation; only for straight-upward traveling radiation do we have that $\beta_\downarrow = b$, whereas $\beta_\downarrow > b$ for all other zenith angles. From geometrical arguments, $\beta_\downarrow = \beta_\uparrow$ for isotropic radiation, where $\beta_\uparrow$ is the fraction of scattered downwelling radiation that is scattered upwards, known as the upscatter fraction $\beta_\uparrow$[54]. For isotropic radiation, this upscatter fraction depends only on the phase function $P$, which defines the probability distribution of the scattering angle $\theta$ relative to the direction of propagation. Neglecting multiple scattering interactions, Wiscombe and Grams[52] showed that

$$\beta_\uparrow = \frac{1}{2\pi}\int_0^\pi \theta P(\cos\theta)\sin\theta \, d\theta, \quad (5)$$

which we use to calculate $\beta_\downarrow$ (Supplementary Fig. 8). The downscatter fraction thus depends only on the phase function, which we calculate using Mie theory and the refractive indices reported in Supplementary Table 4. The downscatter fraction equals 0.5 in the limit of $D \ll \lambda$ (asymmetry factor $g = 0$) and decreases to zero in the limit of $D \gg \lambda$ ($g = 1$)[52,54]. The upwelling spectral irradiance above the dust layer is affected by absorption and emission by the colder atmosphere above (Supplementary Fig. 1), such that the upwelling spectral irradiance at TOA is reduced:

$$F_{\text{TOA}\uparrow}(\lambda) = [1 - \epsilon_{\text{abv}}(\lambda)]F_{d\uparrow}(\lambda) + \epsilon_{\text{abv}}(\lambda)\pi B(T_{\text{abv}}, \lambda) \quad (6)$$

We then obtain the spectral LW DRE at the TOA by subtracting from Eq. (6) the corresponding equation without the dust layer present

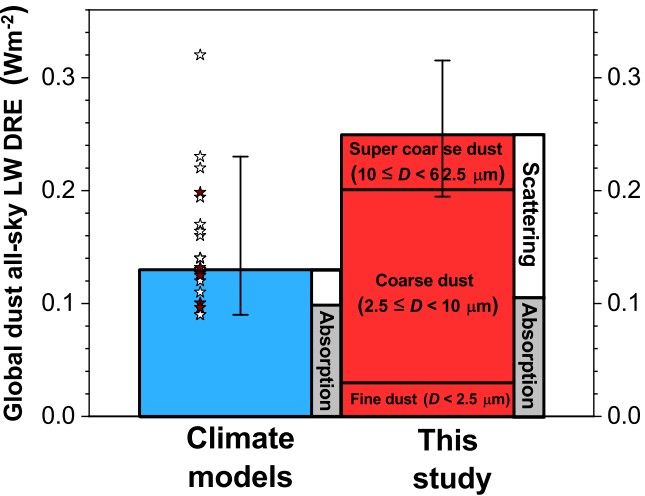

**Fig. 4 | Climate models underestimate the global mean direct radiative effect (DRE) due to dust interactions with longwave (LW) radiation.** A compilation of climate model results shows a global all-sky LW DRE at top-of-atmosphere (TOA) of 0.13 (0.09 – 0.23) Wm⁻² [rendered as Wm$^{-2}$], which is primarily due to LW absorption (gray vertical box) with a small contribution from LW scattering (small white vertical box; Supplementary Table 1). In contrast, our data-driven analytical model constrains the all-sky LW DRE to almost double that value, 0.25 ± 0.06 Wm⁻², more than half of which is generated by LW scattering interactions. These interactions are omitted by most global models, which contributes to the underestimation by a factor of approximately 2 by those models. Additionally, the majority of the LW DRE is generated by coarse and super coarse dust, which is underestimated by global models[17]. Filled brown stars denote global model results in the DustCOMM ensemble (see Fig. 2b and Supplementary Table 1), open stars denote published model results, and error bars represent 90% confidence intervals.

($\tau_{\text{LW}} \to 0$) and rearranging terms:

$$\Delta F_{TOA\uparrow}(\lambda) = -\pi\left[1 - \epsilon_{\text{abv}}(\lambda)\right]\left\{\epsilon_d(\lambda)\left[B\left(T_{s,\text{eff}}, \lambda\right) - B(T_d, \lambda)\right]\right.$$
$$\left. + R_d(\lambda)\left[B\left(T_{s,\text{eff}}, \lambda\right) - \epsilon_{\text{abv}}(\lambda)B(T_{\text{abv}}, \lambda)\right]\right\}, \quad (7)$$

The first term in Eq. (7) represents the effect of absorption, which is mitigated by emission at $T_d$; the second term represents the effect of downward scattering of upwelling radiation by dust, the effect of which is mitigated somewhat by upward scattering of downwelling atmospheric radiation. The integration of Eq. (7) over the full LW spectrum then yields the LW DRE at the TOA:

$$R_{\text{TOA}} = -\int \Delta F_{\text{TOA}\uparrow}(\lambda)d\lambda = \int \pi\left[1 - \epsilon_{\text{abv}}(\lambda)\right]\left\{\epsilon_d(\lambda)\left[B\left(T_{s,\text{eff}}, \lambda\right) - B(T_d, \lambda)\right]\right.$$
$$\left. + R_d(\lambda)\left[B\left(T_{s,\text{eff}}, \lambda\right) - \epsilon_{\text{abv}}(\lambda)B(T_{\text{abv}}, \lambda)\right]\right\}d\lambda, \quad (8)$$

where a minus sign was added because a decrease of the outgoing LW radiation corresponds to a gain of energy to the climate system and thus a positive LW DRE.

Equations (7) and (8) show explicitly that the dust LW DRE at TOA decreases with absorption above the dust layer. Dust radiative effects are therefore negligible for wavelengths for which the atmosphere is opaque[10]. This leads to two important conclusions: (i) dust radiative effects are only important in the spectral region of the atmosphere that is transparent to LW radiation – the so-called "atmospheric window" around 8 – 14 μm wavelength – because strong absorption by water vapor and other gases make the atmosphere opaque outside of this window, and (ii) the LW DRE at TOA is negligible when clouds – which block the atmospheric window by absorbing strongly and broadly in the LW spectrum – are present above the dust layer (with the exception of optically thin cirrus clouds)[10].

We use the observation that dust radiative effects are only important within the atmospheric window to simplify Eq. (8) by using values of the dust optical properties ($\bar{\omega}$ and $\bar{\beta}_\downarrow$) averaged over the atmospheric window wavelength range (Supplementary Table 4). This is further justified by the uncertainty in these parameters being of similar order of magnitude as their variation in the atmospheric window (see, e.g., Fig. 12 in Ref. 14.). We similarly also use the wavelength-averaged values of the atmospheric absorptivities ($\bar{\epsilon}_{bel}$ and $\bar{\epsilon}_{abv}$) and calculate the LW extinction ($\bar{\tau}_{LW}$, $\bar{\epsilon}_d$, and $\bar{R}_d$) based on the size-resolved column loading and optical properties ($k_{ext}$, $\bar{\omega}$, and $\bar{\beta}_\downarrow$) representative of the entire atmospheric window wavelength range (Supplementary Table 4). That is,

$$R_{TOA} = \pi\left(1 - \bar{\epsilon}_{abv}\right) \int_{\lambda_{min}}^{\lambda_{max}} \left\{ \bar{\epsilon}_d \left[ B\left(T_{s,eff}, \lambda\right) - B\left(T_d, \lambda\right)\right] \right. \\ \left. + \bar{R}_d \left[ B\left(T_{s,eff}, \lambda\right) - \bar{\epsilon}_{abv} B\left(T_{abv}, \lambda\right)\right] \right\} d\lambda, \quad (9)$$

We now simplify Eq. (9) further by evaluating the integral of the Planck function over the atmospheric window. We use the Stefan-Boltzmann law to write

$$\int_{\lambda_{min}}^{\lambda_{max}} \pi B(T_{emit}) d\lambda = \sigma_{SB} f_{aw}(T_{emit}) T_{emit}^4, \quad (10)$$

where $\sigma_{SB}$ is the Stefan-Boltzmann constant and $f_{aw}$ is the fraction of emitted radiation that is in the atmospheric window, which is a weakly increasing function of the emitting temperature ($T_{emit}$) at the range of temperatures encountered in the troposphere (Supplementary Fig. 13). Substituting Eq. (10) into Eq. (9) finally yields the clear-sky LW DRE at TOA produced by dust in an atmospheric column:

$$R_{CS} = \sigma_{SB}\left(1 - \bar{\epsilon}_{abv}\right) f_{aw}\left(T_{s,eff}\right) T_{s,eff}^4 \left[\bar{\epsilon}_d \left(1 - \frac{f_{aw}(T_d)}{f_{aw}\left(T_{s,eff}\right)} \frac{T_d^4}{T_{s,eff}^4}\right)\right. \\ \left. + \bar{R}_d \left(1 - \bar{\epsilon}_{abv} \frac{f_{aw}(T_{abv})}{f_{aw}\left(T_{s,eff}\right)} \frac{T_{abv}^4}{T_{s,eff}^4}\right)\right]. \quad (11)$$

Equation (11) shows that the TOA LW DRE has two distinct contributions. The first contribution (left-hand term inside the square brackets) is due to dust absorption of radiation that is emitted from the warmer surface and atmosphere below. The radiative effect of this absorption is countered by the emission of LW radiation by the dust layer at a lower temperature. As such, this term depends on the temperature difference of the dust layer with the surface and atmosphere below, which in turn is largely controlled by the height of the dust layer. The second contribution (right-hand term inside the square brackets) is due to the downward scattering of upwelling LW radiation by dust. This contribution is countered somewhat by upward scattering of downwelling radiation emitted by the overlying atmosphere. This causes a weaker dependence on dust layer height than occurs for LW absorption, such that the relative importance of LW scattering increases with decreasing dust layer altitude[10]. Note that the contributions of both LW absorption and LW scattering to the TOA LW DRE are decreased by the absorption and emission of LW radiation by the colder atmosphere above the dust layer.

**Using dust optical properties, DustCOMM, and reanalysis data to calculate LW DRE at TOA during clear-sky conditions**

We want to use Eq. (11) to constrain the climatology of the LW DRE at TOA, as a function of longitude, latitude, and time (season). Doing so requires quantification of all the variables and their uncertainties in Eq. (11), starting with the dust optical properties. We obtained the downscatter fraction ($\bar{\beta}_\downarrow$), single-scattering albedo ($\bar{\omega}$), and the mass extinction efficiency ($\bar{k}_{ext}$), which co-determines the dust aerosol optical depth ($\bar{\tau}_{LW}$), from Mie theory using six different data sets of published LW optical properties (see Supplementary Methods for details). This yielded values of $\bar{\beta}_\downarrow$ that decrease from 0.5 for very fine dust to ~ 0.15 for super coarse dust, values of $\bar{\omega}$ that increase strongly with particle diameter from ~ 0 for very fine dust to ~ 0.5 for super coarse dust, and values of $\bar{k}_{ext}$ ranging from ~ 0.08 to 0.2 m²g⁻¹ (see Supplementary Methods and Supplementary Table 4). Since the dust size distribution is variable in space and time, so are the corresponding bulk dust optical properties (Supplementary Fig. 8).

The second ingredient needed to use Eq. (11) is the spatiotemporal pattern of the size-resolved dust concentration, which co-determines $\bar{\tau}_d$ and $T_d$. We obtained this from the DustCOMM data set[1], which constrained the climatology of the size-resolved concentration of dust as a function of latitude, longitude, height, and season from observational and modeling constraints on dust properties and abundance for the years 2004–2008. The DustCOMM data up to a diameter of 20 μm is based on Adebiyi and Kok[17], which obtained the spatially and seasonally resolved dust size distribution that minimizes the disagreement against a compilation of in situ dust size distribution measurements. This study also obtained error bounds on the dust size distribution, which are propagated into the uncertainties on the results reported here using the bootstrap procedure (see Supplementary Methods). We extended this data set to include dust with diameters between 20 to 100 μm using simulations from Meng et al.[31] of the ratio of dust mass loading in this size range with dust mass loading for particles with $D \le 20$ μm. As described in more detail in Adebiyi et al.[18], in order to optimally match in situ measurements of super coarse dust size distributions far from source regions, these simulations used a dust density reduced by a factor of 10 (250 kg m⁻³) as a proxy for as-of-yet unclear processes missing from models that likely cause coarse dust to deposit less quickly than simulated in models[31]. These simulations indicate that dust with $D > 20$ μm accounts for ~ 2–4% of the global mean LW DAOD (ref. 18 and Supplementary Table 2), although in situ measurements suggest that this might be an underestimation[32]. As these simulations estimate that the contribution of giant dust ($D > 62.5$ μm) to the LW DAOD is < 0.1%[18], we do not diagnose this fraction separately, but include its contribution in our results for the super-coarse ($10 < D < 62.5$ μm) fraction.

The final ingredient needed to use Eq. (11) is data on surface properties (temperature and emissivity) and atmospheric properties (vertical profiles of temperature and absorptivity and the downwelling radiation at the surface), which co-determine $\bar{\epsilon}_{abv}$, $\bar{\epsilon}_{bel}$, $T_{bel}$, $T_{abv}$, and $T_d$. We obtained surface temperature ($T_s$) from the MERRA-2[23] meteorological reanalysis data set (Supplementary Fig. 12a). Furthermore, we assumed that ocean surface emissivity is 0.985 based on theory and observations[55] and obtained land surface emissivity from the five wavelength bands of land surface emissivity retrieved by the Advanced Spaceborne Thermal Emission and Reflection Radiometer (ASTER) in the atmospheric window[56]. Note that some deserts have surface emissivity substantially less than 1 (Supplementary Fig. 12b), particularly the Sahara desert, which is important to account for in accurate calculations of the LW DRE[57]. Finally, data on atmospheric absorptivity and downwelling radiation at the surface were obtained for clear-sky conditions, averaged over the atmospheric window (using $\lambda_{min} = 8$ μm to $\lambda_{max} = 14$ μm) by forcing the LibRadTran radiative transfer model[58,59] with MERRA-2[23] seasonally averaged reanalysis data of 2D surface temperature and 3D atmospheric temperature, atmospheric humidity and ozone. We obtained these seasonally averaged data for 6-hour increments (0, 6, 12, and 18 UTC) to account for the effect of diurnal variability in surface temperature and in vertical profiles of atmospheric water vapor, ozone, and temperature on the dust LW DRE. Relative to using diurnally averaged data, the effect of this accounting for diurnal variability was of the order of a few percent over land and less over ocean. As such, using higher temporal

resolution data would have had a negligible impact on our results compared to other uncertainties in the analysis. All 6-hourly and seasonally averaged reanalysis data was further averaged over the years 2004–2008 to match the period for which the DustCOMM dust climatology data was obtained[1].

Combining all these ingredients together yielded the spatiotemporal pattern of the LW DRE at TOA for clear-sky conditions (Fig. 1).

## Using model simulations to calculate all-sky LW DRE from clear-sky LW DRE

The approach above constrains the clear-sky LW DRE (Figs. 2, 3a), but the all-sky LW DRE is more important for the Earth's energy balance. If clouds are present above the dust layer then the TOA LW DRE is essentially zero[10,20], as is likely also the case for the SW DRE[60]. However, if clouds are present below the dust layer, then these normally decrease the effective surface temperature $T_b$, thereby somewhat decreasing the TOA LW DRE while increasing the fraction of that LW DRE that is due to scattering. These interactions are too complicated to account for in our analytical model and we thus use results from climate models to convert the clear-sky to the all-sky LW DRE at TOA:

$$R_{AS}(s, \theta, \phi) = \eta(s, \theta, \phi) R_{CS}(s, \theta, \phi), \tag{12}$$

where $s$ denotes the season, $\theta$ and $\phi$ denote longitude and latitude, and $\eta$ is the spatiotemporally varying ratio of the all-sky to the clear-sky LW DRE at TOA. We use results from the six different models in our ensemble (see Supplementary Fig. 6) and propagate the uncertainty due to the spread in these results using a bootstrap procedure (see Supplementary Methods). Note that a limitation of Eq. (12) is that these model simulations of $\eta$ do not account for the effect of dust LW scattering.

## Propagation of uncertainty and use of LW DREE observations using bootstrap procedure

Each of the data sets used in the calculation of the LW DRE has uncertainties, which we propagated to the extent possible using a bootstrap method that also integrates observationally based estimates of the LW DREE. In order to compare our results against these observations, we used our analytical model to calculate the LW DREE ($\Omega_{mdl}$) by dividing the clear-sky LW DRE ($R_{CS}$, Eq. 11) by the clear-sky dust aerosol optical depth in the shortwave spectrum at 550 nm ($\tau_{SW}$). That is,

$$\Omega_{mdl} = \frac{R_{TOA}}{\tau_{SW}} = \sigma_{SB}(1 - \bar{\epsilon}_{abv}) f_{aw}(T_{s,eff}) T_{s,eff}^4 \left[ \frac{\bar{\epsilon}_d}{\tau_{SW}} \left( 1 - \frac{f_{aw}(T_d)}{f_{aw}(T_{s,eff})} \frac{T_d^4}{T_{s,eff}^4} \right) \right. \\ \left. + \frac{\bar{R}_d}{\tau_{SW}} \left( 1 - \bar{\epsilon}_{abv} \frac{f_{aw}(T_{abv})}{f_{aw}(T_{s,eff})} \frac{T_{abv}^4}{T_{s,eff}^4} \right) \right]. \tag{13}$$

Because both $\bar{\epsilon}_d$ and $\bar{R}_d$ scale with $\bar{\tau}_{LW}$, a major determinant of $\Omega_{mdl}$ is the ratio of the clear-sky LW to SW DAOD ($\bar{\tau}_{LW}/\tau_{SW}$), which is plotted in Supplementary Fig. 14.

We performed a sufficiently large number of simulations (1000) that our results did not change substantially with additional simulations. For each simulation, we drew from the probability distributions or ensembles of the data sets that are inputs to the analytical model (see Supplementary Fig. 2). We then compared the calculated LW DREE (Eq. 13) against the compilation of observational estimates (see below) and, in a procedure similar to that used in perturbed parameter ensembles[61], we only retained the simulations consistent with these observational estimates (Supplementary Fig. 3). However, this procedure is hindered by the fact that most LW DREE observations did not include uncertainties, and even those studies that did[57,62] accounted for different factors in this uncertainty. Therefore, we estimated a common uncertainty on all reported observational DREE values as the standard deviation of groups of LW DREE values for similar regions. Specifically, the various observations of LW DREE over the springtime Sahara (six

total; ref. 22) show a standard deviation of 1.6 Wm$^{-2}$; observations of LW DREE over the summertime Sahara (eight total; refs. 22,57,63.) show a standard deviation of 2.5 Wm$^{-2}$; and the two measurements over the tropical North Atlantic in September[64,65] show a spread of 3.3 Wm$^{-2}$. Based on this, we estimate an observational error of ± 2 Wm$^{-2}$. Accordingly, simulations that perfectly reproduce nature would be expected to have a root-mean-squared error (RMSE) of RMSE$_{min}$ = ~2 Wm$^{-2}$ relative to these observations. And indeed, simulations in our bootstrap ensemble have a minimum RMSE of ~ 2 Wm$^{-2}$, so similar to RMSE$_{min}$ (Supplementary Fig. 3). We therefore retained simulations with twice this minimal error, so with RMSE < RMSE$_{max}$, where RMSE$_{max}$ = 2 × RMSE$_{min}$ = 4 Wm$^{-2}$. This procedure eliminated the ~ 55% of the bootstrap simulations that are in the poorest agreement with the LW DREE observations. The result that nearly half of our bootstrap iterations are statistically consistent with the compilation of LW DREE observations supports closure between the "bottom-up" calculation of the LW DRE and "top-down" constraints from in situ and satellite data. However, this closure could be due to canceling errors (see Supplementary Methods). Note that our main results are relatively insensitive to the exact value of RMSE$_{max}$. In fact, applying no constraint (RMSE$_{max}$ = ∞) yields a median all-sky LW DRE of + 0.25 Wm$^{-2}$, which is identical to our results using RMSE$_{max}$ = 4 Wm$^{-2}$ (Fig. 4). Moreover, using an RMSE$_{max}$ of 3 Wm$^{-2}$, which retains only ~ 24% of bootstrap iterations, also yields a similar median all-sky LW DRE of + 0.27 Wm$^{-2}$.

The bootstrap procedure yields a probability distribution of the dust LW DRE, which we use to quantify the errors in our results[66,67]. An analysis of the LW DRE resulting from different bootstrap simulations shows that the uncertainty on the LW DRE arises primarily from uncertainty in the LW refractive index (Supplementary Fig. 3), followed by variability in the global model simulations used to invert the global dust cycle in the DustCOMM data set[1], which co-determine the dust spatial distribution (including dust height). Uncertainty due to the dust size distribution plays a somewhat smaller role. Overall, the uncertainties obtained through the bootstrap procedure should be seen as a lower bound because of the possibility of systematic errors that were not accounted for, including in the observational LW DREE estimates. These and other limitations, as well as the bootstrap procedure, are described in more detail in the Supplementary Methods.

## Compilation of observational estimates of the clear-sky LW DREE

Over a dozen studies have used observations to estimate the clear-sky LW DREE. Those studies can be roughly divided into two groups. The first group of studies used ground-based and/or in situ measurements of radiative fluxes, dust aerosol properties (e.g., size distribution), and/or atmospheric and surface properties (temperature and humidity profiles) to inform and constrain a radiative transfer model that was then used to calculate the clear-sky LW DREE[62,68]. The second group of studies combined satellite remote sensing data of SW (dust) aerosol optical depth and LW flux measurements to estimate the clear-sky LW DREE, often also using a radiative transfer model[22,24]. We combined estimates of both types of studies into a compilation of observational estimates of the clear-sky LW DREE at TOA. For quality control purposes, we excluded studies that (i) did not account for the effect on TOA LW fluxes of the co-variability of dust with atmospheric humidity and surface temperature[69–71], which confound the effects of dust on OLR in the atmospheric window[22] (ii) studies that did not use observations of LW fluxes to constrain the results from a radiative transfer model[72], and (iii) studies that were based on very small amounts of observational data or had very large uncertainties in those data[73]. We did not include the results of Brindley[74] because these results were superseded by Brindley and Russell[22]. We also did not include the results of Kuwano et al.[75] because these results, obtained near the Salton Sea in California, are not representative of the long-range transported dust that is most relevant to climate, but rather of locally emitted dust confined within 1-2 km of

the ground, which models struggle to accurately represent in marginal source regions like the U.S. Southwest[76]. Furthermore, we combined the estimates of di Sarra et al.[77] and Meloni et al.[78], both of which were obtained during spring time at the Mediterranean island of Lampedusa and had similar methods and author teams. Overall, we identified 11 studies that met these criteria, yielding a total of 21 observationally informed estimates of the clear-sky LW DREE (see Supplementary Table 5). To directly compare the observational clear-sky LW DREE estimates to the seasonally and diurnally averaged results presented in this paper, we applied a correction factor to convert all observations in our compilation to a seasonally and diurnally averaged value (see Supplementary Methods).

### Limitations of the methodology

Our methodology is subject to limitations, which we discuss in detail in the Supplementary Methods and summarize below. First, simplifying assumptions and treatments required to keep the analytical model solvable can cause biases in our results. These include assuming that dust and atmospheric absorption are small (i.e., $\bar{\tau}_{LW}$, $\bar{\epsilon}_{bel}$, $\bar{\epsilon}_{abv} \ll 1$), using atmospheric and dust properties spectrally averaged over the atmospheric window rather than resolving spectral variability, and using seasonally averaged inputs, thereby neglecting sub-seasonal co-variability between input fields. Second, the conversion of the clear-sky LW DRE calculated by the data-driven analytical model to the all-sky LW DRE most relevant for Earth's climate and energy balance relied on an ensemble of model simulations of the ratio of the clear-sky to the all-sky LW DRE, which is likely to have biases because these models do not include LW scattering by dust and also struggle to accurately simulate the properties and spatiotemporal distribution of both dust and clouds. Third, our results could be biased due to errors in the input data (e.g., atmospheric temperature and humidity fields and dust concentration, size distribution, altitude, and optical properties). Fourth, errors in observational LW DREE estimates could cause biases. These observational estimates often focus on intense dust events with atypical properties, may conflate surface temperature responses with radiative effects, and are regionally and seasonally biased towards North African dust in the boreal spring and summer seasons, with few observations for boreal fall and none for boreal winter or for any season in the Southern Hemisphere (Fig. 1). As such, systematic differences in dust properties with season and location that are not captured by the analytical model could induce a bias in our results. These limitations are partially mitigated by retaining only bootstrap iterations consistent with observational constraints (Supplementary Fig. 3), but substantial biases remain possible because compensating errors could still result in agreement with these observations. The uncertainties on our results should therefore be interpreted as lower bounds.

## Data availability

The data shown in Figs. 1, 2 and 3 have been deposited in the Zenodo database under accession code 18880560. The data for Fig. 4 are provided in Supplementary Tables 1 and 2.

## Code availability

The code used to conduct the analysis presented in this paper and in the production of the figures is available from https://doi.org/10.24433/CO.5409430.v1.

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

## Acknowledgements

This work was developed with support from the National Science Foundation (NSF) grants 1856389 and 2151093 awarded to J.F.K. A.K.G. was supported by NSF grant 2151093 during his research tenure at UCLA. We also acknowledge high-performance computing support from NCAR's Computational and Information Systems Laboratory, sponsored by NSF. A.I. acknowledges MEXT-Program for the advanced studies of climate change projection (SENTAN) grant number JPMXD0722681344. L.L. and N.M.M. acknowledge assistance from the Earth surface Mineral dust source InvesTigation (EMIT), a NASA Earth Ventures-Instrument (EVI-4) Mission, as well as from the Department of Energy (DOE) under award DE-SC0021302. A.A.A. acknowledges support from the U.S. Department of Energy (DOE), Office of Science (award DE-SC0024281). M.K. has received funding from the Helmholtz Association's Initiative and Networking Fund (grant agreement no. VH-NG-1533). C.P.G.-P. and V.O. acknowledge support from the Spanish Ministerio de Economía y Competitividad through the HEAVY project (grant no. PID2022-140365OB-I00 funded by MCIN/AEI/10.13039/501100011033 and by ERDF/EU), the ERC under the Horizon 2020 research and innovation program through the FRAGMENT project (grant agreement no. 773051), the Horizon Europe program under grant agreement no. 101137680 via project CERTAINTY, and the AXA Research Fund through the AXA Chair on Sand and Dust Storms at BSC. M.K. and C.P.G.-P. acknowledge PRACE for granting access to MareNostrum at the Barcelona Supercomputing Center to run MONARCH. R.L.M. received support from the NASA Modeling, Analysis and Prediction Program. P.R.C. and A.R.L received support from the NASA Atmospheric Composition: Modeling and Analysis Program and the NASA Center for Climate Simulation (NCCS) for computational resources.

## Author contributions

J.F.K. conceived the project, designed the study, performed the analysis, and wrote the paper. A.K.G. contributed simulations of atmospheric absorptivity, and A.T.E. contributed to designing the data-driven analytical model of the dust longwave direct radiative effect. A.A. and Y.H. provided dust optical properties data. S.A., Y.B., R.C.-G., P.R.C., D.H., A.I., M.K., L.L., N.M.M., R.L.M, V.O., C.P.G.-P., A.R.L., and J.W. contributed global model simulation data. All authors discussed the results and commented on the manuscript.

## Competing interests

The authors declare no competing interests.
