## [Transparent Peer Review file · Nature Communications]

Desert dust exerts twice the longwave radiative heating estimated by climate models

Corresponding Author: Professor Jasper Kok

Version 0:

Reviewer comments:

Reviewer #2

(Remarks to the Author)

I see that the authors have addressed my previous comments and reflected them in the revised manuscript by adding discussions about the model and data uncertainty. I think the manuscript is much more rigorous. I only have two remaining minor comments.

1. The regional and seasonal representativeness of the results should be better noted, since (1) they are validated mainly using data from North Africa and (2) for boreal winter and fall, there is almost no validating observation, but the southern hemisphere still has substantial dust DREE.
2. Could the authors separately examine dust longwave scattering effect for day and night? The authors indicate that omitting this effect may lead to overestimated daytime cooling and underestimated nighttime heating, so a quantitative estimate could be more enlightening, especially for climate implications.

(Remarks on code availability)

Reviewer #3

(Remarks to the Author)

This article deals with the long under-examined, yet important topic of the longwave direct radiative effect of mineral dust on a global scale. Recent progress around improved quantification of dust size distributions and refractive indices, both of which are important for the direct longwave radiative effect (DRE), make the study particularly timely. The authors apply a data-driven analytical model using an impressive variety of well-chosen methods, data sources and theories to determine dust's longwave DRE. They validate their findings using available observations constraining the DRE. They find the LW DRE is nearly twice that found in conventional climate models – attributing this to inaccurate size distributions and omitted longwave scattering – a significant difference and omission which will need to be tackled if climate models are to be able to represent dust realistically.

The authors have responded to remarks from reviewer #1 and made substantial adjustments to the manuscript. In particular, results of dust radiative forcings have been removed with the emphasis now on dust radiative effect (i.e. radiative effect of the presence of dust in the present climate), split into shortwave and longwave contributions, as well as total. The structure and content of the manuscript as they appear now are relevant, significant and clear.

The methodology and data quality, though complex, are sound and clearly support the conclusions. The manuscript has already undergone revisions at this stage. I found it to be of high quality and very clear. Based on the responses to reviewer 1, and the manuscript in this latest form, I recommend the article is accepted for publication subject to minor amendments listed below.

Specific Points

Part of the response to reviewer 1 includes removing LW radiative forcing results. However please check there are no lingering mentions of radiative forcing (e.g. L264, and further mentions in the discussion). These discussions are still relevant, but I suspect a clear differentiation between DRE and DRF has been lost in revisions. It would be good to explicitly state the meaning of DRE and DRF at some point. L303 also refers to forcing where it looks like it should be DRE.

Table S1 – Woodward et al. (2022) includes LW DRE TOA estimates also. Also Woodage & Woodward (2014). These also include LW scattering. Their values vary between being within the range cited on L75 and sitting above this range. Is there a reason these studies were omitted?

Extended data fig 1 – does not have much text or explanation around it, either in the main article or extended methods. Extended methods seems a suitable location to explain further and also link to the description of DRE equations.

L144-5 – ‘capture less than a quarter of the variance..’ – what exactly is meant by this, as the variability in fig 2b for the models looks similar to the observations. Same point for line 190-1 (‘models inadequately capture the spatiotemporal variability of LW DREs’)

L160-161- I think it would be clearer here to use the abbreviation ‘LWDREE’ and include the ‘per tau_sw’ in the units to avoid confusion with the DRE itself.

L166 – ‘low’ cloud cover – change to ‘small’ or similar to avoid confusion with low altitude clouds.

L239, Fig 4 – citing Table S2 for climate models’ contribution of scattering to LW DRE – the cross-reference must be incorrect because table S2 only contains data from the analytical model.

L247-249 – it is not clear how Fig S8 demonstrates this statement. Which sub-plot of S8 is being referred to? (or all?) The magnitudes of values in the third column appear large and spatially varied. Please clarify.

I was surprised not to see more discussion around models which *do* include LW scattering from dust, as some indicated in table S1 do, and as the studies I indicate above do as well. Do they produce different results? By how much? Are they in line with or different from this study’s analytical model values?

L271 – ‘underestimating LW heating biases...’ If referring to atmospheric heating rates, this should be ‘cooling’ not heating. Or is TOA warming meant here? Clarification is needed.

L269-274 – I think this section generally refers to the radiative effects of dust in a regime where dust overlies stratocumulus cloud – this should be specified.

L436-7 – ‘dust with $D > 20$ microns accounts for ~2% of the global mean LW DAOD’ – this seems to contradict the data in table S2?

L556 and below – Limitations of methodology – the rather simplistic method of calculating the all-sky DRE from the clear-sky DRE should also be included here.

Why do reference numbers start at #7 in the introduction, where are 1-6? (Maybe lost in revisions?)

Supplement

S6 – how is the downscatter fraction calculated?

Fig S1 – bubble stating ‘Analytical model (Eqs 12 & 13)’ – please check that is correct? It looks like these equations deal with the all-sky calculations, not clear-sky DRE as indicated in the flow diagram.

References

Woodward et al., 2022, ACP - The simulation of mineral dust in the United Kingdom Earth System Model UKESM1, <https://doi.org/10.5194/acp-22-14503-2022>

Woodage and Woodward, 2014, U.K. HiGEM: Impacts of Desert Dust Radiative Forcing in a High-Resolution Atmospheric GCM in: Journal of Climate Volume 27 Issue 15 (2014), <https://doi.org/10.1175/JCLI-D-13-00556.1>

(Remarks on code availability)

Version 1:

Reviewer comments:

Reviewer #3

(Remarks to the Author)

The authors have now addressed all reviewer comments credibly, adding alterations, clarifications and corrections to the manuscript where necessary.

My previous statement about the significance and rigour of the article still stand.

I therefore suggest the manuscript is accepted for publication.

(Remarks on code availability)

Other than checking the link works, I have not investigated the code further.

We thank the reviewers for their careful reading of the paper and their constructive comments. Below, we describe how we addressed each comment in blue.

Reviewer #2 (Remarks to the Author):

I see that the authors have addressed my previous comments and reflected them in the revised manuscript by adding discussions about the model and data uncertainty. I think the manuscript is much more rigorous. I only have two remaining minor comments.

1. The regional and seasonal representativeness of the results should be better noted, since (1) they are validated mainly using data from North Africa and (2) for boreal winter and fall, there is almost no validating observation, but the southern hemisphere still has substantial dust DREE.

The reviewer is correct that the observational data compiled in Fig. 1 is mainly for North Africa and mainly for boreal spring and summer, with few observations for boreal fall and none for boreal winter or the Southern Hemisphere. We have expanded the text in the Methodology sub-section "Limitations of the methodology" to more clearly point this out (newly added text is underlined):

"These observational estimates often focus on intense dust events with atypical properties, may conflate surface temperature responses with radiative effects, and are regionally and seasonally biased towards North African dust in the boreal spring and summer seasons, with few observations for boreal fall and none for boreal winter or for any season in the Southern Hemisphere (Fig. 1). As such, systematic differences in dust properties with season and location that are not captured by the analytical model could induce a bias in our results."

2. Could the authors separately examine dust longwave scattering effect for day and night? The authors indicate that omitting this effect may lead to overestimated daytime cooling and underestimated nighttime heating, so a quantitative estimate could be more enlightening, especially for climate implications.

That is a great question. Unfortunately, our analysis is not set up to obtain an answer to this question because it uses 6-hourly reanalysis data at set UTC times (0, 6, 12, and 24 hours). Since the daytime and nighttime hours thus vary strongly with longitude (as well as with latitude and season), it would require substantial additional effort to calculate the nighttime versus daytime cooling.

However, it's important to note that, compared to the SW radiative effects, the LW radiative effects of dust have a relatively small diurnal cycle, with the difference between

radiative effects centered around noon and those averaged over a full day being approximately 20% (Table S5). Therefore, the overestimate of dust's daytime surface cooling and underestimate of its nighttime surface heating is caused simply by underestimating (the diurnally quasi-constant) LW heating effects and is not primarily due to diurnal variation in these LW heating effects.

To make this clearer in the main text, we have adjusted the relevant sentence to (added text underlined):

“Although the atmospheric response to these radiative perturbations is mediated by interactions with the large-scale circulation³, the net effect of underestimating dust LW radiative effects – whose magnitude varies only weakly over the diurnal cycle – is to overestimate dust's daytime surface cooling while underestimating its nighttime surface heating.”

Reviewer #3 (Remarks to the Author):

This article deals with the long under-examined, yet important topic of the longwave direct radiative effect of mineral dust on a global scale. Recent progress around improved quantification of dust size distributions and refractive indices, both of which are important for the direct longwave radiative effect (DRE), make the study particularly timely. The authors apply a data-driven analytical model using an impressive variety of well-chosen methods, data sources and theories to determine dust's longwave DRE. They validate their findings using available observations constraining the DREE. They find the LW DRE is nearly twice that found in conventional climate models – attributing this to inaccurate size distributions and omitted longwave scattering – a significant difference and omission which will need to be tackled if climate models are to be able to represent dust realistically.

The authors have responded to remarks from reviewer #1 and made substantial adjustments to the manuscript. In particular, results of dust radiative forcings have been removed with the emphasis now on dust radiative effect (i.e. radiative effect of the presence of dust in the present climate), split into shortwave and longwave contributions, as well as total. The structure and content of the manuscript as they appear now are relevant, significant and clear.

The methodology and data quality, though complex, are sound and clearly support the conclusions. The manuscript has already undergone revisions at this stage. I found it to be of high quality and very clear. Based on the responses to reviewer 1, and the manuscript in this latest form, I recommend the article is accepted for publication subject to minor amendments listed below.

We thank the reviewer for their careful reading of the paper and for their positive and constructive comments.

Specific Points

Part of the response to reviewer 1 includes removing LW radiative forcing results. However please check there are no lingering mentions of radiative forcing (e.g. L264, and further mentions in the discussion). These discussions are still relevant, but I suspect a clear differentiation between DRE and DRF has been lost in revisions. It would be good to explicitly state the meaning of DRE and DRF at some point. L303 also refers to forcing where it looks like it should be DRE.

Thank you for pointing this out. We have corrected the two occurrences on L264 and L303 and have also searched for any other such occurrences but have found none. As suggested, we have also now added a clear definition of the direct radiative forcing (new text underlined):

Our constraints on the LW DRE also provide a critical step toward resolving the sign and magnitude of the net dust direct radiative forcing –the change in the dust direct radiative effect from pre-industrial to present-day - and its contribution to climate change.

Table S1 – Woodward et al. (2022) includes LW DRE TOA estimates also. Also Woodage & Woodward (2014). These also include LW scattering. Their values vary between being within the range cited on L75 and sitting above this range. Is there a reason these studies were omitted?

Thank you for pointing this out as we indeed overlooked these two studies in our compilation. We have now added these to Table S1 and have updated the text and Figure 4 accordingly.

Extended data fig 1 – does not have much text or explanation around it, either in the main article or extended methods. Extended methods seems a suitable location to explain further and also link to the description of DRE equations.

To add visibility to this conceptual figure, we have added additional references to it in the main text.

L144-5 – ‘capture less than a quarter of the variance..’ – what exactly is meant by this,

as the variability in fig 2b for the models looks similar to the observations. Same point for line 190-1 ('models inadequately capture the spatiotemporal variability of LW DREs')

The variance is the square of the standard deviation and R^2 denotes the fraction of variance explained. We've now made that link between R^2 and the variance explained clearer by noting the average of the R^2 values for the models:

In contrast, global model simulations underestimate the LW DREE by approximately a factor of 2 and, on average, explain less than a quarter of the variance in observations (average $R^2 = 23\%$; Fig. 2b, Table S3).

The spatiotemporal variability refers to the variance in the LW DREE observations, which vary in time. To clarify this, we've adjusted the wording to "spatiotemporal variance", which is more consistent with the wording on L144-5 and mathematically precise.

L160-161- I think it would be clearer here to use the abbreviation 'LWDREE' and include the 'per tau_sw' in the units to avoid confusion with the DRE itself.

Thank you for this suggestion - we have corrected this accordingly.

L166 – 'low' cloud cover – change to 'small' or similar to avoid confusion with low altitude clouds.

Thank you for pointing that out as that phrasing is indeed confusing. We have changed it to "sparse cloud cover".

L239, Fig 4 – citing Table S2 for climate models' contribution of scattering to LW DRE – the cross-reference must be incorrect because table S2 only contains data from the analytical model.

This indeed should be Table S1. Corrected accordingly.

L247-249 – it is not clear how Fig S8 demonstrates this statement. Which sub-plot of S8 is being referred to? (or all?) The magnitudes of values in the third column appear large and spatially varied. Please clarify.

We have now clarified that the first mention of Fig. S8 (L. 250) refers to the left column of figures and that the second mention (L255) refers to the right column.

I was surprised not to see more discussion around models which *do* include LW

scattering from dust, as some indicated in table S1 do, and as the studies I indicate above do as well. Do they produce different results? By how much? Are they in line with or different from this study's analytical model values?

That's a good question. The reason for this is two-fold. First, the studies that the reviewer pointed out include the effects of LW scattering but did not investigate what the contribution of LW scattering to their simulated LW radiative effects was. Because of large differences between model simulations (e.g., because of differences in optics and dust size distribution) we therefore cannot do more than speculate what the effect of LW scattering in those simulations was. Second, the other studies in Table S1 that list a contribution from LW scattering obtained this not by actually simulating LW scattering, but rather by simply scaling their simulated LW radiative effects (due to only absorption) by some factor. For these reasons, we see little scientific value in comparing our results to those from these models.

L271 – 'underestimating LW heating biases...' If referring to atmospheric heating rates, this should be 'cooling' not heating. Or is TOA warming meant here? Clarification is needed.

L269-274 – I think this section generally refers to the radiative effects of dust in a regime where dust overlies stratocumulus cloud – this should be specified.

Thanks for these helpful observations. This is indeed referring to dust overlying stratocumulus clouds, for which LW radiative effects heat the cloud tops. We've changed this sentence to make this clearer:

“Moreover, because ~~dust~~ absorption and emission of LW radiation by dust overlying stratocumulus clouds ~~modulate~~ reduce cloud-top cooling and modulate boundary-layer stability, underestimating LW ~~heating~~ radiative effects biases the representation of dust semi-direct effects on stratocumulus clouds⁴⁵.”

L436-7 – 'dust with D>20 microns accounts for ~2% of the global mean LW DAOD' – this seems to contradict the data in table S2?

That's an astute observation. The results in the cited reference are specifically for the DAOD at 10 um wavelength, whereas the results in Table S2 are the average for the 8-14 um spectral range. We've now adjusted the sentence to give a range indicated by both results, referring to both sources:

“These simulations indicate that dust with $D > 20 \mu\text{m}$ accounts for $\sim 2\text{-}4\%$ of the global mean LW DAOD (Ref. ¹⁹ and Table S2), although in situ measurements suggest that this might be an underestimation³⁴.”

L556 and below – Limitations of methodology – the rather simplistic method of calculating the all-sky DRE from the clear-sky DRE should also be included here.

That’s indeed an important limitation that should be mentioned. We have accordingly added the following sentence:

“Second, the conversion of the clear-sky LW DRE calculated by the data-driven analytical model to the all-sky LW DRE most relevant for Earth’s climate and energy balance relied on an ensemble of model simulations of the ratio of the clear-sky to the all-sky LW DRE, which is likely to have biases because these models do not include LW scattering by dust and also struggle to accurately simulate the properties and spatiotemporal distribution of both dust and clouds.”

Why do reference numbers start at #7 in the introduction, where are 1-6? (Maybe lost in revisions?)

Good catch. That was due to referencing not having been updated after some edits to the introduction. This has been corrected and the references now start at 1.

Supplement

S6 – how is the downscatter fraction calculated?

The downscatter fraction was calculated as reported right below in Eq. S8, and numerical values are reported in Table S4. We also note in the next paragraph that these calculations were performed using Mie theory.

Fig S1 – bubble stating ‘Analytical model (Eqs 12 & 13)’ – please check that is correct? It looks like these equations deal with the all-sky calculations, not clear-sky DRE as indicated in the flow diagram.

Thank you for pointing this out – this indeed should refer to Eqs. 11 and 13. We’ve corrected the figure accordingly.

References

Woodward et al., 2022, ACP - The simulation of mineral dust in the United Kingdom Earth System Model UKESM1, <https://doi.org/10.5194/acp-22-14503-2022>

Woodage and Woodward, 2014, U.K. HiGEM: Impacts of Desert Dust Radiative Forcing in a High-Resolution Atmospheric GCM in: Journal of Climate Volume 27 Issue 15 (2014), <https://doi.org/10.1175/JCLI-D-13-00556.1>